# Role of Artificial Intelligence for Autism Diagnosis Using DTI and fMRI: A Survey

**DOI:** 10.3390/biomedicines11071858

**Published:** 2023-06-29

**Authors:** Eman Helmy, Ahmed Elnakib, Yaser ElNakieb, Mohamed Khudri, Mostafa Abdelrahim, Jawad Yousaf, Mohammed Ghazal, Sohail Contractor, Gregory Neal Barnes, Ayman El-Baz

**Affiliations:** 1Department of Diagnostic Radiology, Faculty of Medicine, Mansoura University, Elgomheryia Street, Mansoura 3512, Egypt; emanata_1978@yahoo.com; 2Bioengineering Department, University of Louisville, Louisville, KY 40292, USA; aaelna02@louisville.edu (A.E.); yaser.elnakieb@utsouthwestern.edu (Y.E.); mskhud02@louisville.edu (M.K.); maabde02@louisville.edu (M.A.); 3Electrical, Computer, and Biomedical Engineering Department, Abu Dhabi University, Abu Dhabi 59911, United Arab Emirates; jawad.yousaf@adu.ac.ae (J.Y.); mohammed.ghazal@adu.ac.ae (M.G.); 4Department of Radiology, University of Louisville, Louisville, KY 40202, USA; sohail.contractor@louisville.edu; 5Department of Neurology, Pediatric Research Institute, University of Louisville, Louisville, KY 40202, USA; gregory.barnes@louisville.eduu

**Keywords:** autism spectrum disorder (ASD), fMRI, DTI, artificial intelligence, deep learning, survey, diagnostics

## Abstract

Autism spectrum disorder (ASD) is a wide range of diseases characterized by difficulties with social skills, repetitive activities, speech, and nonverbal communication. The Centers for Disease Control (CDC) estimates that 1 in 44 American children currently suffer from ASD. The current gold standard for ASD diagnosis is based on behavior observational tests by clinicians, which suffer from being subjective and time-consuming and afford only late detection (a child must have a mental age of at least two to apply for an observation report). Alternatively, brain imaging—more specifically, magnetic resonance imaging (MRI)—has proven its ability to assist in fast, objective, and early ASD diagnosis and detection. With the recent advances in artificial intelligence (AI) and machine learning (ML) techniques, sufficient tools have been developed for both automated ASD diagnosis and early detection. More recently, the development of deep learning (DL), a young subfield of AI based on artificial neural networks (ANNs), has successfully enabled the processing of brain MRI data with improved ASD diagnostic abilities. This survey focuses on the role of AI in autism diagnostics and detection based on two basic MRI modalities: diffusion tensor imaging (DTI) and functional MRI (fMRI). In addition, the survey outlines the basic findings of DTI and fMRI in autism. Furthermore, recent techniques for ASD detection using DTI and fMRI are summarized and discussed. Finally, emerging tendencies are described. The results of this study show how useful AI is for early, subjective ASD detection and diagnosis. More AI solutions that have the potential to be used in healthcare settings will be introduced in the future.

## 1. Introduction

Autism spectrum disorder (ASD) is a long-term neurodevelopmental disorder characterized by impaired social communication and interaction, restricted and repetitive stereotypical behavior patterns, and diminished cognitive skills. The World Health Organization (WHO) estimates that ASD affects about 67 million individuals around the world. Males are four times more affected than females [1,2,3]. The exact etiology of ASD is still unclear. Heterogeneous and multi-factorial causes are suggested, including genetic background [4].

Autistic symptoms usually begin to develop within the first two years of life. Early ASD manifestations can be found in 12-month-old infants. However, the average age for diagnosis is around five years [1,5]. The current gold standard in ASD diagnosis is based on behavior observational tests by clinicians such as the Autism Diagnostic Observation Schedule (ADOS) or Autism Diagnostic Interview-Revised (ADI-R) report, but these approaches are subjective and time-consuming. Early and appropriate diagnosis is crucial to help limit the deterioration of the condition and to improve prognostic outcomes [2,6,7].

Magnetic resonance imaging (MRI) is an essential non-invasive method in the detection of brain structure, white-matter (WM) integrity, and functional activity [8]. Structural MRI (sMRI) has been used to describe the morphological brain changes in ASD regarding the shape and volume of different brain regions. Diffusion tensor imaging (DTI) provides an assessment of anatomical connections and has shown disorganized micro-structural WM integrity in the autistic population. Functional MRI (fMRI) relies on the detection of dynamic physiological information from active brain regions. Measuring the change in blood-oxygenation-level-dependent (BOLD) signals in various brain states (resting state or task-evoked) can reveal functional architecture abnormalities in the ASD population [9,10]. Although different MRI modalities have shown promise in distinguishing ASD individuals from healthy controls (HCs), MRI results remain inconsistent and nonreplicable [8]. Therefore, the need for neuroimaging biomarkers remains an ongoing clinical challenge. Several computer-aided design systems (CADS) have been widely applied to integrate multimodal MRI with artificial intelligence (AI). Machine learning (ML) is a subfield within (AI). In neuroimaging, ML is widely used in medical image analysis through extracting informative features and constructing the best-fitting algorithm to provide the desired output [5,11]. The most frequently selected features for ASD include color, shape, texture, and spatial relationship features. These features are computed to study developmental brain abnormalities and can be applied to improve the diagnosis and to classify subtypes and the degree of severity of ASD [12,13].

The availability of large datasets, including those from the Autism Brain Imaging Data Exchange (ABIDE), has led to an increase in publications combining ML with different neuroimaging biomarkers. These studies aim to reduce subjectivity and to establish a more objective, data-driven method for identification, classification, and prognosis of ASD children [5].

In this survey, we aim to review the publications predicting or identifying ASD based on different MRI modalities (i.e., DTI and fMRI) using ML methods. DTI studies since 2011 are presented. There has been a great increase in the number of publications based on functional imaging (fMRI). Here, we focus on recent fMRI studies in the last five years. A manual search is done using electronic databases in PubMed and Google Scholar for articles and papers published in English until July 2022 using search terms as follows: (autism, autism spectrum disorders, or ASD) and (diffusion tensor imaging or DTI) AND (functional magnetic resonance imaging, fMRI, task-based fMRI, T-fMRI, resting state fMRI, rs-fMR, or BOLD) and (artificial intelligence, AI, machine learning, ML, deep learning, or DL) and (detection, diagnosis, or findings). The eligibility criteria included original research articles published, accepted for publication, or available online in English. Age- or sex-based studies were included. Case reports and review articles, including narrative, systematic reviews, and meta-analyses, were excluded from data extraction but were used as reference searches. Studies comparing a group of ASD individuals with a group of typically developed controls were included. However, studies conducted on the comparison of ASD with other neurodevelopmental, cognitive, or psychiatric disorders such as attention-deficit hyperactive disorder (ADHD) were excluded. Given that this review is designed to look at DTI and fMRI studies based on ML findings, ML algorithms with neuroimaging data were used as a biomarker in differentiating ASD individuals from typically developed controls. In addition, other imaging modalities such as structural MRI, MR spectroscopy, or positron emission tomography were excluded as well.

## 2. MRI Findings for ASD

### 2.1. ASD Findings Using Diffusion Tensor Imaging (DTI)

DTI is a non-invasive, in vivo tool that measures water diffusion within WM tracts, thus providing a macroscopic picture of WM ultrastructure within the imaged voxel. The most important DTI metrics are mean diffusivity (MD) and fractional anisotropy (FA). MD measures the overall amount of diffusion and is related to cellular density. FA captures the directional changes of diffusion and represents the degree of alignment of WM tracts and cellular structure, ranging from 0 (random or isotropic) to 1 (unidirectional or anisotropic). Other parameters include axial diffusivity (AD) and radial diffusivity (RD). AD measures the diffusion in a direction parallel to WM tracts and represents axon integrity, whereas RD is the perpendicular diffusion and is related to myelin integrity [14,15,16].

White-matter tracts are composed of bundles of axons that carry the communication signals between brain regions. Alteration in synaptogenesis caused by dysmaturation of myelination has been reported in the ASD population. Myelin alteration leads to changes in axonal fiber density, caliber, and homogeneity with subsequent impairment of WM microstructural organization and integrity [10,17,18].

Many studies have shown reduced FA and increased MD in widespread WM tracts and brain regions of the ASD population when compared to typically developed controls (TD), denoting reduced WM integrity. The most-commonly involved tracts are long-range association fibers that directly and indirectly connect the brain regions responsible for social cognition and verbal communication. Those tracts include the superior longitudinal fasciculus, occipitofrontal fasciculus, arcuate fasciculus, uncinate fasciculus, inferior longitudinal fasciculus, and cingulum [1,19,20,21]. A study by Jung et al. [19] revealed the correlation between impaired connectivity at the occipital cortex in ASD boys with the core symptoms and clinical outcomes of ASD. A significant negative correlation was found between tract length (left cingulum cingulate gyrus and right uncinate fasciculus) and the total score of the Social Communication Questionnaire (SCQ).

Valenti [22] presented a comprehensive review of several published articles that used DTI in the evaluation of corpus callosum (CC) integrity in ASD. They found a significant difference in DTI and tractography findings between the ASD group and healthy controls, indicating both micro- and macro-structure alterations in CC. In addition, those structural alterations are correlated with socio–communicative deficits. Shukla et al. [23] found increased RD and reduced FA in CC and the internal capsule (IC). Others have shown altered IC connectivity with the correlation of DTI changes with the core ASD symptoms [24,25,26]. A study focused on language-related tracts (arcuate fasciculus) to differentiate ASD patients from non-ASD individuals with a developmental language disorder found a significant reduction in FA of the arcuate fasciculus in ASD individuals [27].

Moreover, other studies reported age-related differences in the widespread WM micro-structure of ASD population when compared to healthy controls. For example, in [28,29], the authors found increased FA in autistic infants and toddlers, while indices decreased in elder autistic children. This was attributed to better tract coherence and alignment in infancy. Other studies have shown the opposite, as there was a significant positive correlation of FA with increasing age of autistic children while MD and RD measures showed a significant negative correlation with age [30,31]. These studies suggest that neurodevelopmental maturation of WM trajectories with increased age is associated with changes in diffusivity parameters.

Apart from using DTI biomarkers alone, several imaging-based ML studies have been applied to overcome the limitations found in DTI studies alone. DTI lacks a full description of crossing WM trajectories. Despite the sensitivity of DTI metrics to capture microstructural changes, DTI is less specific for other WM disorders affecting myelination and axonal density [32]. In addition, there is still limited integration between clinical and imaging biomarkers. The need for informative data relevant to diagnosis and treatment decisions is challenging. Therefore, ML has been developed to aid the identification and classification of ASD children using clinical, behavioral, and imaging biomarkers [10].

### 2.2. ASD Findings Using fMRI

Functional neuroimaging is used to investigate the functional connectivity and activity of brain regions. Electroencephalography (EEG) has been used as a basic method to record electrical activities of the brain from the scalp with high temporal resolution (in milliseconds) [33]. Further, fMRI detects brain activity by measuring the associated variations in blood-oxygenation-level-dependent spontaneous signals (BOLD) in response to various stimuli [34]. It is a four-dimensional technique (4D) that captures the three-dimensional brain volume (3D) repeatedly over a period of time. This technique has high spatial resolution (in millimeters) but low temporal resolution [35]. This can be explained by the time limitations of fMRI, which cannot record the fast dynamics of brain activity and the slow response of the brain hemodynamic system, thus requiring multiple scans over time. Moreover, fMRI is sensitive to motion artifacts [36]. Techniques based on fMRI include two broad categories: event-related or task-based (T-fMRI) and resting-state (rs-fMRI). Task-based (T-fMRI) measures brain function after performing specific tasks, while (rs-fMRI) measures brain function in the absence of task demands.

#### 2.2.1. Task-Based (T-fMRI)

Task-based protocols employ paradigms that map the core behavioral symptoms of ASD patients, namely, facial emotional recognition, response to social stimuli, and reward behavior [1]. Other tasks include motor, visual processing, language, auditory, and executive functions. Social communication skills are supported by a distributed brain network within different brain regions collectively named the “social brain” [37]. Many T-fMRI studies have notably demonstrated atypical activity in social brain regions among ASD individuals during social tasks (primarily hypoactivation).

Social communication deficits in ASD include variable manifestations such as impaired recognition of faces, making inferences about others’ intentions, and diminished social responsiveness. In addition, there is reduced attention to social cues, human voices, and biological motions [1,37,38]. A study that examined the activity of social brain regions in response to the visual perception of generic faces showed hypoactivation in the fusiform gyrus and amygdala among ASD children. These regions are responsible for such tasks. However, similar activation was found in ASD patients as well as TD controls when the faces were familiar [39]. Another study employed a task to distinguish between attention to biological motions (eye gaze, walking, hand, or mouth movements) versus mechanical motions (clock or wheel). This study revealed reduced activation of the superior temporal sulcus and ventrolateral prefrontal cortex in ASD children compared to TD controls. The study clarified that ASD children are easily distracted by non-facial stimuli and cannot fixate on faces to the same degree as normal children [40].

Investigating brain activity in response to cognition of facial expressions, such as sad facies (visual tasks), revealed increased activation in the amygdala, ventral prefrontal cortex, and striatum in the adolescent ASD group compared to the control group [41].

Regarding the response to reward or positive feedback behavior, ASD children are less responsive than normal children. Normally, a reward activates the visual striatum region and engages the frontostriatal network [1]. A study employed a social reward task such as a smiling face or a momentary reward task such as gold coins. ASD boys showed a nonactivated visual striatum, while it was activated in TD boys Scott-Van Zeeland AA. Moreover, a study comparing the response of both ASD boys and girls revealed more activation in the lateral frontal cortex and insula of ASD girls, denoting that suppressed reward center activation is a distinctive feature of ASD boys [42].

A meta-analysis of fMRI studies was proposed by Philip et al. [43]. They reviewed T-fMRI studies that investigated the functional brain response to auditory and language-related tasks. They revealed reduced activation in clusters of brain regions in ASD children, adolescents, and adults compared to TD controls. Those regions are both superior temporal gyri, the right pyramids of the cerebellar vermis, and the left middle cingulate gyrus. The superior temporal gyrus is activated with receptive language, so reduced activation in response to spoken language denotes the underlying verbal communication difficulties in ASD individuals. On the other hand, relative over-activation in ASD adults compared to controls was found in the posterior cingulate gyrus, the motor cortex, and the cerebellar declive. The exact cause of increased activation in ASD groups was not clear; this may suggest the use of atypical language processing strategies [43].

#### 2.2.2. Resting-State (rs-fMRI)

The complexity and different varieties of fMRI tasks in addition to the unique social skills and intellectual condition of ASD children can limit some task-based experiments. Another point is the difficulty of tasks with potential language barriers that preclude some children, particularly infants, from participation [1,44].

Resting-state (rs-fMRI) is a promising alternative to T-fMRI. The technique is suitable for infants and toddlers. It helps examine functional brain connectivity in the absence of task performance. The total time of the examination is about 5–6 min. The participants just lie in the MRI scanner with their eyes closed or with their vision fixed on a crosshair [44]. Several rs-fMRI studies aim to explore large-scale resting brain networks (RBNs). These are organized brain regions that show cortical synchronization patterns with coherent spontaneous fluctuations in neural activity during rest [45]. RBNs include the default mode network, dorsal and ventral attention network, salient network, visual network, and sensorimotor networks [45].

It has been found that the most common region with altered brain connectivity is the default mode network (DMN). It becomes activated during the resting state, whereas it becomes less activated with the engagement of cognitively-demanding tasks [46]. DMN is a large-scale network composed mainly of the posterior cingulate gyrus, precuneus, and medial prefrontal cortex. It has shown reduced connectivity in ASD patients compared with TD controls [46,47,48]. The dorsal attention network (DAN) is located in the intraparietal sulcus and frontal eye field. This network is activated to reorient the attention towards relevant stimuli. Sun et al. [45] showed increased functional connectivity in the superficial temporal gyrus and cerebellum, indicating the presence of circuit connections between the DAN and cerebellum. The ventral attention network (VAN) also has been studied by SUN [45]. The authors found increased connectivity in the insula, which is a critical region in VAN responsible for social emotions. The salience network is another RSN that has been examined. It is composed primarily of the dorsal anterior cingulate cortex and the anterior insula. It is linked with the detection and filtering of salient stimuli [1]. Uddin et al. [49] observed a reduction in functional connectivity in this network with 83% accuracy for differentiating autistic children from TD controls. Additionally, a study by Wang et al. [50] investigated the functional connectivity of sensory networks in autistic children, including auditory, visual, and sensorimotor networks. They found increased functional connectivity in all networks in ASD children that was correlated with the severity of social impairment of the children.

Several other networks have been examined by rs-fMRI and have provided support to the theory of “brain hypoconnectivity” in ASD. Long-range reduced brain connectivity was found in the superior temporal region of autistic individuals when compared with TD controls [51]. In addition, other studies have observed underconnectivity in the latero–basal subregion of the amygdala, interhemispheric connectivity in the sensorimotor and occipital cortices, and underconnectivity in connections between the anterior and posterior cingulate gyrus and the precuneus [52]. Overconnectivity has been also observed in some studies; researchers found increased connectivity in some areas, such as the frontal, temporal, and occipital regions [48,53,54]. Kleinhans and his colleagues [55] observed areas of overconnectivity within the amygdala (superficial and centro–medial subregions). However, the latero–basal subregion showed underconnectivity; this region stands for the presentation and severity of ASD symptoms. The mixed pattern or the inconsistent over- and underconnectivity can be attributed to the small sample size used in these studies, phenotypic heterogeneity among ASD individuals, or an adjustment mechanism by the brain to bypass the underconnected regions [1,44].

Overall, both DTI and fMRI have revealed important findings that are associated with ASD. These findings are summarized in Figure 1.

## 3. The Role of AI in ASD Diagnosis

Artificial intelligence (AI) involves the replication of human thinking and problem-solving using artificially intelligent components. Machine learning (ML) is an integral aspect of AI that involves utilizing image processing tools to extract features from an input database. The data are then categorized through unsupervised learning or classified into grades through supervised learning. Supervised learning uses labeled input–output pairs to classify data, with classifiers such as SVM, random forest, and traditional neural networks being commonly used. Deep learning (DL), a subset of ML, has become increasingly popular in the medical field. The most commonly employed deep learning networks are convolutional neural networks (CNNs). They consist of many convolutional and fully connected layers to perform feature extraction and classification. Conversely, unsupervised learning categorizes data based on input data patterns without the need for labeled input–output pairs.

In modern times, AI has become a significant factor in numerous applications, including the early identification and diagnosis of ASD (see Figure 2). This paper briefly reviews various AI-based techniques for the early detection and diagnosis of ASD. To evaluate the effectiveness of AI components, diverse metrics are utilized to address medical concerns such as the classification, diagnosis, and early detection of eye diseases. A short summary of these metrics is provided below. True positive (TP), true negative (TN), false negative (FN), and false positive (FP) are represented by the abbreviations. The ensuing performance metrics are defined in the following manner:Specificity: $\frac{TN}{FP+TN}$;Sensitivity (recall): $\frac{TP}{TP+FN}$;Accuracy: $Accuracy=\frac{TP+TN}{TP+TN+FP+FN}$;Precision: $\frac{TP}{TP+FP}$;F1-score: $\frac{2*\mathrm{Precision}*\mathrm{Recall}}{\mathrm{Precision}+\mathrm{Recall}}=\frac{2*TP}{2*TP+FP+FN}$;The AUC is the region beneath the receiver operating characteristics (ROC) curve that shows the relationship between the false positive rate (1-specificity, shown on the *x*-axis) and the true positive rate (sensitivity, on the *y*-axis). The AUC value ranges from 0 to 1, with a higher AUC value indicating better performance.

### 3.1. The Role of AI for ASD Diagnosis Using DTI

Currently, ML has been implemented with DTI metrics to evaluate structural connectivity changes in ASD population. Several ML studies based on DTI metrics have been performed by applying different features obtained using different algorithms for image processing. For example, Ingalhalikar et al. [56] extracted DTI-based features such as FA and MD metrics in each ROI to classify ASD patients and controls by learning the pattern of the disease. Further, they correlated the degree of ASD with the clinical score of each subject to aid in the diagnosis. A nonlinear SVM was used for classification. Li et al. [57] used brain connectivity network features obtained from DTI to identify faulty sub-networks to distinguish ASD subjects from the HC group. The detection was done by an SVM- recursive feature elimination (RFE) algorithm. Jin et al. [58] used features extracted from ROI-based WM connectivity and DTI metrics such as as FA, MD, and fiber length. A multi-kernel SVM was used for the discrimination of ASD 6-month-old high-risk infants from low-risk infants. Zhang et al. [59] used whole brain fiber clustering analysis with multi-fiber tractography, FA, and MD features. An SVM classifier was used to distinguish ASD from HCs. Qin et al. [60] used graph theory to analyze the topology of the white-matter network of ASD preschool children. Edges and nodes were defined as FA and 90 brain regions, respectively. They found disturbed topology of the structural networks of ASD subjects as compared to HCs. Payabvash [32] used features extracted from edge density imaging (EDI) as well as from conventional DTI metrics such as FA, MD, and RD. Variable ML classifiers were used to discriminate ASD children from HCs. The best accuracy was achieved with the EDI-based random forest model. Saad et al. [61] used DTI-based connectivity features with graph theory to classify ASD and HCs. The classification was performed by an SVM and a linear discriminant analysis (LDA). The principle component analysis (PCA) approach was used to reduce noisy features. Better accuracy was obtained with SVM and two PCA features. ElNakieb et al. [62] used global and local extraction of FA, MD, AD, RD, and skewness features to examine the performance of CAD systems in ASD diagnosis. The detection was performed using an SVM classifier. In [63], the authors used global and local extraction of FA, MD, AD, RD, and skewness features to identify significantly different paired WM areas in ASD and then to classify ASD individuals and HCs. They used an SVM model for classification. In [64], the authors also used global and local extraction of FA, MD, AD, RD, and skewness features to classify ASD individuals and HCs. They used linear and nonlinear SVM classifiers. The best classification accuracy was achieved with linear SVM (LSVM).

Other hybrid studies have combined DTI features with features extracted either from functional MRI (fMRI) or structural MRI (sMRI). For example, An et al. [65] used region-to-region fiber connectivity DTI features and ROI-based functional connectivity (FC) features from fMRI to validate connectivity patterns and then classify ASD subgroups using a multi-view expectation maximization formulation (mv-EM). Deshpande et al. [66] used FA and FC values with a multi-variate autoregressive model (MVAR). They investigated the differences between brain regions that may underpin the theory of mind in young ASD patients and HCs. The classification was done by an SVM classifier. Crimi et al. [67] used structural and functional connectivity features with a constrained multivariate auto-regressive model (CMAR) that allows fusing the structural connectivity with the information from the functional time series to represent effective brain connectivity. The classification between ASD subjects and HCs was performed by an SVM classifier. D’Souza et al. [68] used phenotypic measures of rs-MRI connectivity and DTI tractography features with a multimodal graph convolutional network (M-GCN) to extract predictive biomarkers from both ASD individuals and HCs. Irimia et al. [69] used structural morphometric features such as cortical thickness, volume, area, and mean curvature as well as connectivity features to distinguish ASD individuals from HCs. Moreover, they distinguished ASD males and females. The used classifier was an SVM model. Eill et al. [70] combined the anatomical features (surface area, mean curvature, cortical thickness, volume, and local gyrification index), DTI metrics (FA, MD, RD, and AD), and ROI-based FC matrices. They applied a conditional random forest algorithm (CRF) to assess the role of each modality and to explore the more informative one in diagnostic prediction. The use of combined variables achieved higher accuracy (92.5%), and rs-fMRI connectivity variables provided better performance than other anatomical modalities in the classification of ASD individuals from HCs. Figure 3 highlights the different extracted features and the most-used classifiers in DTI autism diagnostic systems. Table 1 summarizes ASD studies of DTI with ML models and briefly outlines the dataset used and its demographics, the feature selection, the ML classifier used, findings, and the accuracy whenever reported.

### 3.2. The Role of AI for ASD Diagnosis Using fMRI

Recent progress in ML algorithms combined with fMRI techniques has been established for the diagnosis of ASD and has shown promising results. In this review, we include several high-impact publications from the last five years. For example, Abraham et al. [71] used features selected from ROI-based resting-state FC matrices to differentiate ASD from HCs. They used an SVM classifier and found an increase in the predictive power with the increase in participant numbers. Emerson et al. [72] used extracted features from ROI-based resting-state FC matrices for the prediction of ASD in high-risk 6-month-old infants based on correlated brain metrics with 24-month ASD-related behaviors. They achieved 96.6% classification accuracy of ASD individuals at 24 months. The classification was performed by an SVM classifier. Guo et al. [73] used ROI-based resting-state FC matrices with deep neural networks with feature selection (DNN-FS) to select the most relevant features from FCs related to ASD from the default mode to identify ASD individuals from HCs. Jahedi A [74] obtained ROI-based FC features as biomarkers to identify ASD patients from HCs. Combined use of conditional random forest (CRF) and random forest classifiers achieved the best prediction accuracy. Kam et al. [75] used extracted features from seed-based FC matrices with discriminative restricted Boltzmann machines (DRBM) to identify dominant FCs to differentiate ASD from HCs. An ensemble classifier was applied and obtained high accuracy with multiple clusters using hierarchical-level clustering of networks. Sadeghi et al. [76] used features from ROI-based FC nodal matrices to extract local and global parameters of brain networks to identify ASD from HCs. Multiple classifiers were used, but SVM showed superiority to other classifiers. Subbaraju et al. [77] used extracted features from ROI-based resting-state FC matrices with a spatial-feature-based detection method (SFM) to extract the most-discriminative blood-oxygenation-level-dependent (BOLD) signals. An SVM classifier was used in the classification of ASD individuals versus HCs. Tejwani et al. [78] used extracted features from ROI-based enhanced FC variability across brain regions to distinguish between ASD subjects and HCs. They used SVM, RF, Naïve Bayes, and multi-layer perception algorithms for classification. Heinsfeld et al. [79] used extracted features from ROI-based FC matrices with a DNN algorithm. The proposed method achieved better accuracy than SVM and RF classifiers in classifying ASD and control subjects. Bi et al. [80] used ROI-based FC features to differentiate between autistic individuals and HCs. They applied a random SVM cluster and achieved high classification accuracy.

Furthermore, Fredo et al. [81] used features extracted from ROI-based FC matrices to classify ASD individuals and HCs. The detection was done by CRF. Li et al. [82] conducted a two-stage pipeline method composed of DNN and prediction distribution analysis to investigate the accuracy in classifying two datasets of ASD and HC subjects. They extracted features from ROI-based FC matrices from both resting-state and task-fMRI. Bernas et al. [83] extracted temporal neurodynamic fMRI biomarkers for ASD diagnosis with wavelet coherence maps. The detection was done by an SVM classifier. Bhaumik et al. [84] used features extracted from ROI-based FC matrices for the prediction and diagnosis of ASD subjects versus HCs. The detection was done by an SVM and partial least square regression (PLS) algorithms. Dekhil et al. [85] used power spectral densities as extracted features to classify ASD individuals and HCs. An SVM was used and achieved high diagnostic accuracy and prediction of clinical phenotypes. Xiao et al. [86] used the extracted time courses of each subject with NN algorithms. The extracted features were inputted into a stacked autoencoder (SAE) and then into a subsequent softmax classifier for the identification of school-aged ASD children versus HC school-aged children. Yang et al. [87] used features extracted from ROI-based FC matrices to classify ASD and HC subjects. The detection was performed by SVM, LR, and ridge classifiers. They found that accuracy is improved with combined classifiers. Wang et al. [88] used extracted features from ROI-based FC matrices to classify ASD and HC subjects. In [89], they used an SVM recursive feature model to achieve high classification accuracy of ASD and HC individuals on both global and across-site datasets. They also used extracted features from ROI-based FC matrices to identify ASD subjects and HCs. An SVM-recursive feature model and a stacked sparse auto-encoder (SSAE) were used to eliminate some meaningless features to enable the SSAE to extract insightful features. Aghdam et al. [90] used extracted features from fast Fourier transformation with the CNN method in order to classify ASD subjects and HCs. Huang et al. [91] used extracted features from rs-fMRI multiple-group sparse networks. They used an SVM classifier to distinguish ASD individuals. Jun et al. [92] used extracted features from rs-fMRI local functional characteristics based on the estimated likelihood of ROI by hidden Markov models (HMMs) to identify meaningful information for ASD detection. The detection was performed by an SVM classifier. Eslami et al. [93] used extracted features from ROI-based FC matrices with ASD-DiagNet in order to classify ASD and HC subjects, and the proposed method achieved high classification accuracy. Mostafa et al. [94] extracted features from ROI-based FC matrices and then the eigenvalues of the Laplacian matrix of the brain network with a sequential feature selection algorithm. Linear discriminant analysis (LDA) was used to classify ASD and HC subjects. Song et al. [95] used extracted features from community pattern analysis of FC to classify the ASD population and HCs in addition to the prediction of the clinical classes of ASD individuals. The detection was performed by LDA. Spera et al. [96] used extracted features from ROI-based FC matrices to classify ASD and HC subjects. They selected a homogeneous cohort of young ASD males to lessen the impact of confounding factors. The detection was by linear kernel SVM. Tang et al. [97] used extracted features from ROI-based FC from DMN and the whole brain with joint symmetrical non-negative matrix factorization (JSNMF). An SVM classifier was used to classify ASD and HC subjects. Yamagata et al. [98] used extracted features from ROI-based FC matrices with a multivariate ML approach. A sparse logistic regression (SLR) classifier was used to classify pairs of ASD patients and their unaffected siblings from pairs of HCs and their siblings according to the endophenotype.

Recently, Chaitra et al. [99] used extracted features from ROI-based FC matrices to classify ASD patients and controls. They employed a recursive-cluster-elimination SVM algorithm. Fan et al. [100] used extracted features from maps based on estimated likelihood values of ROI by HMM to identify ASD individuals from HCs. The detection was performed by an SVM classifier. Liu et al. [101] employed the elastic network method to extract features from ROI-based FC matrices to distinguish ASD individuals from HCs. They obtained high classification accuracy using an SVM classifier in the automatic diagnosis of ASD compared to LASSO and RR algorithms. Hu et al. [102] utilized extracted features from ROI-based FC matrices with a fully connected neural network (FCNN) model in the classification of the ASD population versus HCs. Sherkatghanad et al. [103] used extracted features from ROI-based FC matrices to classify ASD individuals and HCs using a CNN model. A CNN model is computationally less intensive as it uses fewer parameters than state-of-the-art methods and can be used in prescreening of ASD patients. Thomas et al. [104] used the temporal statistics of rs-fMRI data with 3D-CNN to classify ASD individuals. The classification was also performed by an SVM model on the same dataset. The best classification accuracy obtained by the SVM algorithm was 66%, while 3D-CNN achieved 64%, denoting that 3D-CNN could not learn additional information in classifying ASD and HCs. Jiao et al. [105] utilized extracted features from ROI-based FC matrices with the CapsNET method to classify ASD individuals and HCs. Moreover, they stratified ASD subjects into groups based on distinct FC measures. Liu et al. [106] used the extracted features from ROI-based FC matrices with an elastic network method to classify ASD individuals versus HCs. Liu et al. [107] utilized the extracted dynamic features from ROI-based FC matrices with a multi-task feature selection method. A multi-kernel SVM classifier was used to classify ASD individuals versus HCs. Zhang et al. [108] utilized the rs-fMRI dataset of ASD subjects and HCs with a fast entropy method that included approximate entropy (ApEn) and sample entropy (SampEn). The SVM classifier was used to diagnose ASD. Ronicko et al. [109] used extracted features from ROI-based FC matrices with partial and full correlation methods. Classification of ASD individuals and HCs was built by different models, namely, SVM, RF, Oblique RF, and CNN. Khan et al. [110] analyzed the extracted features from ROI-based FC matrices with a teacher–student-neural-network-based feature selection method. Different classifiers were used in classifying ASD and HC subjects, such as SVM, RF, LR, decision trees, and linear discriminant classifiers.

More recently, Reiter et al. [111] used extracted features from ROI-based FC matrices of ASD subjects and HCs. An RF classifier was used to investigate the effect of heterogeneity of ASD samples on classification accuracy. They found that the most-homogeneous samples achieved better RF classifier performance. Devika and Oruganti [112] utilized extracted data from FC matrices to distinguish ASD and HC subjects. The classification was performed by an SVM model. Ahammed et al. [113] used the extracted features from ROI-based FC matrices with the bag-of-features extraction (BoF) method. The SVM was used as a classifier for identifying ASD and HC subjects. Ahammed et al. [114] used the extracted features from ROI-based FC matrices. They applied a DarkASDNet method to classify ASD and HC subjects. Graña and Silva [115] utilized the extracted features from ROI-based FC matrices of ASD subjects and HCs using nine different classifiers. They explored the impact of choices during building up the ML pipelines on the predictive performance. They found that the selection of some feature extraction methods can strengthen the classifier performance, such as classical principal component analysis (PCA) and factor analysis (FA). Al-Hiyali et al. [116] used the temporal dynamic features from default mode network regions (DMNs) with several deep learning models for the diagnosis and classification of ASD. SVM and K-nearest neighbors (KNN) were used for ASD classification, and KNN achieved the highest classification accuracy. Pominova et al. [117] utilized extracted features of FC matrices and full-size MRI series with a 3D convolutional autoencoder method. To classify ASD and HC subjects, the SVM classifier was used. Yin et al. [118] used extracted features from ROI-based FC matrices with graph theory and autoencoders to distinguish ASD subjects from HCs. SVM, K-nearest neighbor (KNN), and DNN algorithms were used for classification. Chu et al. [119] used extracted features from ROI-based FC network regions with a multi-scale graph convolutional network (GCN) to classify ASD patients and HCs by learning the distinctive FC features. Yang et al. [120] used extracted features from FC matrices to distinguish ASD individuals from HCs. They used different classifiers such as LR, KSVM, DNN, and supervised learning classifiers; among these, KSVM achieved the best classification accuracy. Figure 4 highlights the different extracted features and the most-used classifiers in fMRI autism diagnostic systems. Table 2 summarizes ASD studies of fMRI with ML models and briefly outlines the dataset used and its demographics, the feature selection, the ML classifier used, findings, and the accuracy whenever reported.

## 4. Discussions and Future Trends

The use of artificial intelligence (AI) in medical diagnosis is a rapidly growing field, and there has been increasing interest in using AI for the diagnosis of autism spectrum disorder (ASD). This survey article examines the potential role of AI in ASD diagnosis using diffusion tensor imaging (DTI) and functional magnetic resonance imaging (fMRI). DTI is a neuroimaging technique that can be used to assess the microstructure of white matter in the brain, and fMRI is a technique that measures brain activity by detecting changes in blood flow. Both DTI and fMRI have shown promise in the diagnosis of ASD, as they can provide insights into the neural basis of the disorder. This article reviews the literature on the use of DTI and fMRI in ASD diagnosis and the reported findings, highlighting the different machine-learning techniques that have been employed. We also discuss the strengths and limitations of these techniques and suggest areas for future research.

### 4.1. Summary of ASD Findings Using DTI and fMRI

As summarized in Figure 1, both DTI and fMRI have revealed significant differences in ASD brains. In DTI, altered FA and MD have been reported in different brain regions. In addition, alterations in CC integrity have been reported at both micro- and macro-structural levels. Using fMRI, different reports have used T-fMRI to reveal atypical activities during social tasks and different auditory and language tasks. On the other side, rs-fMRI, being more suitable for infants and toddlers as not involving specific tasks, has been used to report altered brain connectivity in DMN, DAN, and salience networks.

To build more meaningful findings using DTI and fMRI, more reports should be carried out to verify how the age of participants affects these findings. In addition, there is a need to carry out studies that involve large numbers of participants to further verify the reported findings.

### 4.2. Summary of ASD Diagnostic Systems Using DTI and fMRI

Figure 3 and Figure 4 summarize the most-utilized features and classifiers in different reported DTI and fMRI autism diagnostic systems. For DTI, different tractography-based or voxel-based features can be extracted. Tractography-based features include region-to-region fiber connectivity, brain connectivity networks, and fiber lengths. Voxel-based features include FA, MD, RD, AD, and skewness. A limited number of classifiers have been reported in the literature, with SVM being the most frequently used. For fMRI, the ROI-based FC matrices were the most frequently used features. In addition, seed-based, wavelet coherent maps, FFT features, and dynamic FC features have also been investigated. A wide variety of classifiers have been reported using fMRI, including regular ML classifiers, neural networks, autoencoder, CNN, and ensembles of different classifiers.

More diagnostic systems should be investigated to further improve diagnostic accuracy, considering more novel features or the fusion between the reported features. In addition, the recent advances in deep learning methods and explainable AI should be investigated.

### 4.3. Strengths of Using DTI and fMRI for ASD Diagnosis

One of the main strengths of using DTI and fMRI for ASD diagnosis is that these techniques provide non-invasive, objective measures of brain structure and function. They can be used to identify subtle brain abnormalities that may not be visible on structural MRI or other imaging modalities and can provide insights into the neural basis of ASD. Another strength of DTI and fMRI is their ability to capture brain activity in real-time, which allows for the study of dynamic brain processes such as language, attention, and social interaction. This can be particularly useful for understanding the specific brain abnormalities that underlie the core deficits of ASD and for developing targeted interventions. Overall, the use of DTI and fMRI in combination with machine learning techniques has the potential to revolutionize the way ASD is diagnosed and treated by providing a more accurate and personalized approach to care.

### 4.4. Limitations of Using DTI and fMRI for ASD Diagnosis

One limitation of using DTI and fMRI for ASD diagnosis is that these techniques are typically only available at specialized centers and may not be readily accessible to all patients. In addition, these techniques can be time-consuming and costly, which may limit their widespread use. Another limitation is that the findings from DTI and fMRI studies in ASD are often inconsistent, and there is a lack of consensus on the specific brain abnormalities that are associated with the disorder. This may be due in part to the heterogeneous nature of ASD as well as the use of different ML techniques and sample sizes in different studies.

To address these limitations and improve the accuracy of ASD diagnosis using DTI and fMRI, it is important to continue to develop and validate ML techniques and to standardize the acquisition and analysis of neuroimaging data. In addition, future research should focus on identifying the specific brain abnormalities that are most strongly associated with ASD and on developing more precise and predictive biomarkers for the disorder.

Overall, the use of AI for ASD diagnosis using DTI and fMRI shows great promise, but there is still much work to be done in order to fully understand the neural basis of ASD and to develop reliable and accurate diagnostic tools. Further research is needed to validate and optimize these techniques and to determine their clinical utility. More specifically, optimized software prototypes need to be developed and tested on large datasets acquired from different laboratories and institutions before commercialization. The availability and cost of these datasets are considered to be one of the largest potential barriers, especially with the lack of infant- and toddler-related MRI data that are required to prove the ability of these diagnostic-imaging-based systems to provide early autism diagnosis.

## 5. Conclusions

The literature reported significant differences in ASD brains using DTI and fMRI. DTI reported altered FA, MD, and CC integrity in different brain regions. T-fMRI reported atypical activity during social tasks and different auditory and language tasks. Finally, rs-fMRI revealed altered brain connectivity in DMN, DAN, and salience networks. These findings encourage researcher to utilize artificial intelligence (AI) and machine learning (ML) techniques for ASD diagnosis using DTI and fMRI. The application of AI to DTI and fMRI mainly involves two basic steps: feature extraction and classification. This survey basically highlights the different databases, features, and classifiers that are used in different autism diagnostic systems. These systems have shown great promise, but there is still much work to be done in order to fully understand the neural basis of ASD and to develop reliable and accurate diagnostic tools. The strengths of DTI and fMRI include their ability to provide non-invasive, objective measures of brain structure and function and their ability to capture brain activity in real-time. However, these techniques also have limitations, including their availability, cost, and the inconsistent findings from different studies. To address these limitations and improve the accuracy of ASD diagnosis using DTI and fMRI, it is important to continue to develop and validate machine learning (ML) techniques and to standardize the acquisition and analysis of neuroimaging data. In addition, future research should focus on identifying the specific brain abnormalities that are most strongly associated with ASD and on developing more precise and predictive biomarkers for the disorder. This requires a concerted effort from researchers across different disciplines, including neuroscience, psychology, and computer science. Overall, the use of AI in ASD diagnosis has the potential to revolutionize the way the disorder is diagnosed and treated by providing a more accurate and personalized approach to care. This will ultimately lead to more effective and personalized treatment strategies for individuals with ASD and could have a significant impact on the quality of life for individuals with this disorder and their families.

## Figures and Tables

**Figure 1 biomedicines-11-01858-f001:**
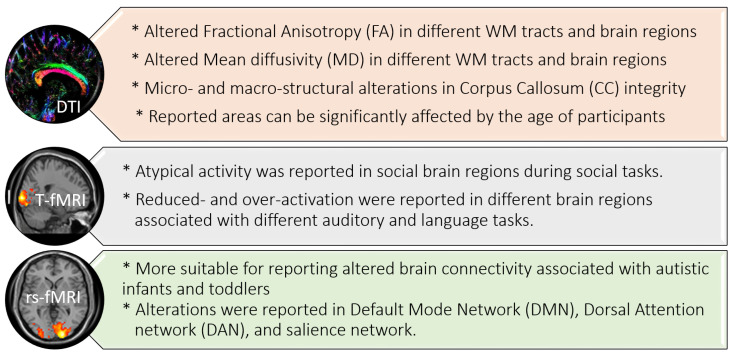
Summary of DTI and fMRI findings in autism.

**Figure 2 biomedicines-11-01858-f002:**
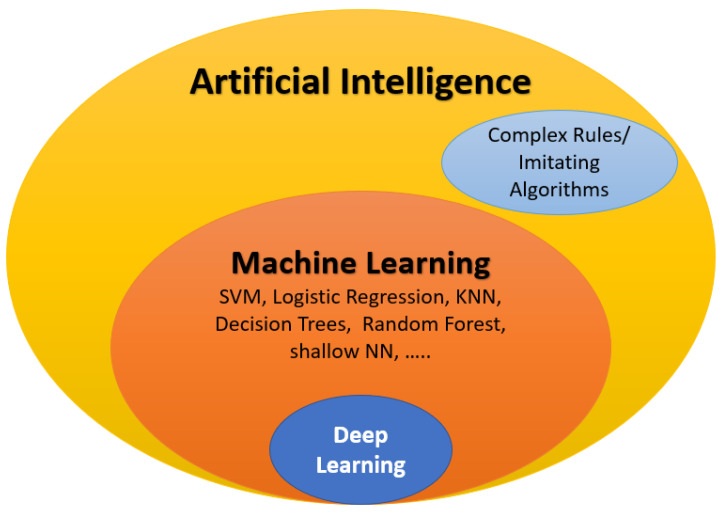
Components of artificial intelligence (AI) and its relation to machine learning (ML) and deep learning.

**Figure 3 biomedicines-11-01858-f003:**
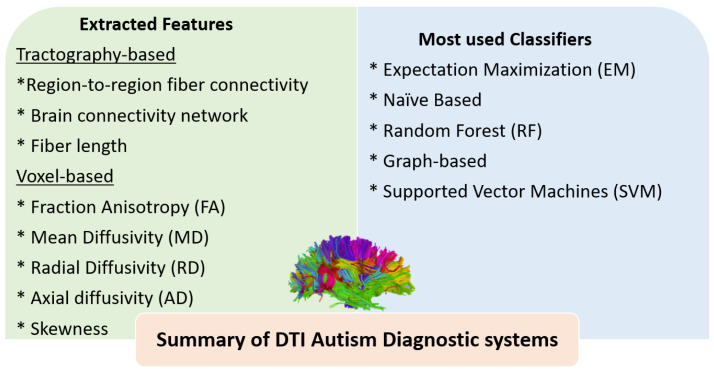
Summary of the most-utilized features and classifiers in different reported DTI autism diagnostic systems.

**Figure 4 biomedicines-11-01858-f004:**
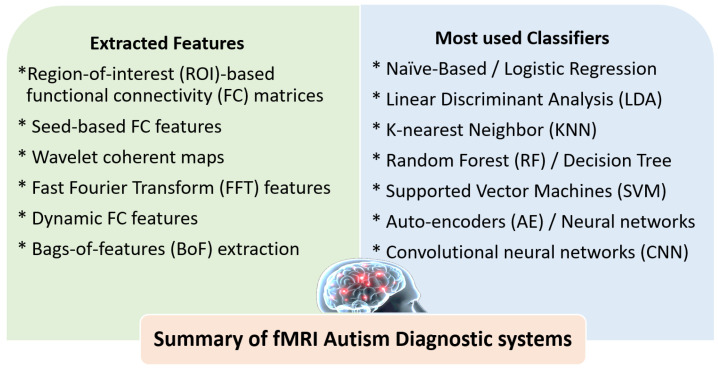
Summary of the most-utilized features and classifiers in different reported fMRI autism diagnostic systems.

**Table 1 biomedicines-11-01858-t001:** Summary of ASD studies of DTI with machine learning (ML) models.

Article	Dataset	ASD	HC	Age	Sex (Male%)	Feature Selection	ML Classifier	Goal/Findings	Accuracy
An et al., 2010 DTI+ fMRI [65]	Clinical	n.s.	n.s.	Child-control dataset	n.s.	DTI: Region-to-region fiber connectivity. fMRI: ROI-based functional connectivity (FC) dataset.	Multi-view expectation maximization (mv-EM) formulation fMRI: ROI-based functional connectivity (FC) dataset (mv-EM).	Goal: Validate connectivity patterns and classify ASD subgroups using mv-EM. Conclusion: Determination of sub-networks of connectivity based on functional and fiber connectivity information.	Classification error: 8.55%
Ingalhalikar et al., 2011 [56]	Clinical	45	30	10.5 ± 2.5	74.6%	FA, MD	SVM	Goal: Identification of ASD individuals versus HC group. Correlate the degree of ASD with clinical score to each subject to aid in the diagnosis. Conclusion: – FA significant difference in ASD group in right occipital region, left superior longitudinal fasciculus, external, and internal capsules. – MD significant difference in right occipital gyrus and right temporal WM.	80%
Li et al., 2012 [57]	Clinical and simulated	10	10	7–14	n.s.	Brain connectivity network	SVM-recursive feature elimination (RFE)	Goal: Identify the faulty sub-networks to distinguish ASD subjects from HC group. Conclusion: Faulty sub-networks can be used as neuroimaging biomarker for computer-assisted diagnosis of ASD.	100%
Deshpande et al., 2013 DTI + fMRI [66]	Clinical	15	15	21.1 ± 0.9	n.s.	FA values. FC values.	Multi-variate autoregressive model (MVAR). SVM.	Goal: Investigate differences between brain regions that may underpin the theory of mind in young ASD patients and HCs. Conclusion: Impaired connectivity in the social brain of ASD group.	95.9%
Jin et al., 2015 [58]	NDAR	40 High-risk infants	40 Low-risk infants	6-month-old infants	70%	FA, MD, and fiber length	Multikernel SVM	Goal: Use whole-brain WM connectivity networks to identify high-risk ASD infants. Conclusion: Identification of potential imaging connectomic biomarkers using multi-parameter multi-scale networks for ASD diagnosis and prognosis of 6-month-old high-risk infants.	76%
Crimi et al., 2017 sMRI+ DTI [67]	ABIDE-II	31	23	n.s.	n.s.	Structural and FC matrix	SVM	Goal: Use a constrained multivariate autoregressive model (CMAR) that allows fusion of the structural connectivity with the information from the functional time series to represent effective brain connectivity. Conclusion: Estimated effective connectivity revealed different brain architecture in ASD group supported by both structural and functional connectivities.	Structural connectivity: 60.57%. Functional connectivity: 72.32%
Zhang et al., 2018 [59]	Clinical	70	79	11.0 ± 2.6	100%	WM fiber clusters-FA, MD	SVM	Goal: Use whole-brain fiber clustering analysis with multi-fiber tractography model for group classification of ASD. Conclusion: FA significantly affected in ASD group; local long tracts are more affected	78.3%
Irimia et al., 2018 sMRI+ DTI [69]	Clinical	110	83	12.74	50%	Structural: Cortical thickness, volume, area, and mean curvature. Connectomic: Connectivity density.	SVM	Goal: Investigate the performance of SVM in distinguishing ASD individuals from HCs as well as ASD males and females. Conclusion: Distinguish ASD from HC; sensitivity = 97.17% – Distinguish ASD males and females; sensitivity = 96.3% – Correlation of volumetric, morphometric, and connectomic with ASD.	94.82%
Qin et al., 2018 [60]	Clinical	39	19	2.89 ± 0.97	82%	Graph-theory-based features	Edges and nodes	Goal: Examine the white-matter connections in the brains of preschool-aged ASD children. Conclusion: – Disturbed topology of the structural networks of ASD subjects. – Nodes in the left precuneus, thalamus, and superior parietal cortex on both sides were found to be more efficient. – The greater the severity of ASD, the greater the nodal efficiency of the left precuneus.	—
Payabvash 2019 [32]	Clinical	14	33	8.9 ± 2.7	100%	Edge density imaging (EDI). FA, MD, and RD.	Naïve Bayes RF SVM	Goal: Distinguish ASD children from HCs using edge density imaging (EDI) based on structural connectivity. Conclusion: Significant lower EDI in ASD; no significant difference in FA, MD, or RD	75.3% in EDI using RF
Saad et al., 2019 [61]	USC Multimodal Connectivity Database	51	41	n.s.	n.s.	Graph-theory-based features	SVM	Goal: Classify ASD and HCs based on graph-theory-based features. Conclusion: Graph-theory-based classification of ASD and HC detected global and local connectivity effectively. Principle component analysis (PCA) approach reduced noisy features for better accuracy.	75%
Eill et al., 2019 aMRI+ DTI+ fMRI [70]	Clinical	46	47	13.63 ± 2.81	84.8%	Cortical characteristics including surface area, mean curvature, cortical thickness, volume, and local gyrification. Index-DTI metrics: (FA, MD, AD, and RD). ROI: based on FC matrices.	Conditional random forest (CRF)	Goal: Classify ASD individuals and HCs based on CRF using a combination of in-house datasets including aMRI, DTI, and fMRI data to assess the role of each modality and explore the more-informative ones in diagnostic prediction. Conclusion: – Resting-state functional connectivity variables provided better performance than other anatomical modalities in the classification of ASD individuals from HCs. – Principle component analysis (PCA) approach reduced noisy features for better accuracy.	For each modality separately: 67%. For combined variables: 92.5%.
ElNakieb et al., 2019 [62]	NADR	122	141	8–17.9	50%	Global and local extraction of FA, MD, AD, RD, and skewness features	SVM	Goal: Examine the performance of CAD system in ASD diagnosis based on WM connectivity. Conclusion: – High global diagnostic decision of ASD by CAD system. Correlation between autistic behavior and DTI changes. – Principle component analysis (PCA) approach reduced noisy features for better accuracy. Scalable system.	71%
ElNakieb et al., 2020 [63]	NADR	124	139	8–17.9	50%	Global and local extraction of FA, MD, AD, RD, and skewness features	SVM	Goal: Investigate the accuracy of CAD system in ASD diagnosis based on WM connectivity. Conclusion: High diagnostic accuracy. Identification of significantly different paired WM areas in ASD. Scalable system.	73%
ElNakieb et al., 2021 [64]	ABIDE-II	125	100	5.1–46.6	n.s.	Global and local extraction of FA, MD, AD, RD, and skewness features	Linear and non-linear classifiers	Goal: Classify ASD individuals using ML based on WM connectivity. Correlation between autistic behavior and DTI changes. Conclusion: Identification of significantly different paired WM areas in ASD.	99%
D’Souza et al., 2021 DTI + rs-fMRI [68]	Clinical	57	275	n.s.	n.s.	Phenotypic measures of rs-MRI connectivity. DTI tractography	Multimodal graph convolutional network (M-GCN)	Goal: Extract predictive biomarkers for ASD diagnosis. Conclusion: – The suggested approach obtains indicative biomarkers from both individuals with no health issues and those with autism. – M-GCN combines structural and functional mapping information to derive phenotypic measures even when training data are limited	—

**Table 2 biomedicines-11-01858-t002:** Summary of ASD studies of fMRI with machine learning (ML) models.

Article	Dataset	ASD	HC	Age	Sex (Male%)	Feature Selection	ML Classifier	Goal/Findings	Accuracy
Abraham et al., 2017 rs-fMRI [71]	ABIDE	403	468	n.s.	83.5%	ROI-based FC matrices.	SVM	Goal: Extract predictive biomarkers to differentiate ASD from HC. Conclusion: Increased predictive power with the increase in participant numbers.	67%
Emerson et al., 2017 [72] fcMRI	Clinical	11 high-risk infants	48	6–24 months high-risk	69.5%	ROI-based FC matrices.	SVM	Goal: The ability to predict ASD in high-risk infants at 6 months and accurately diagnose ASD at 24 months. Conclusion: Early brain metrics found at 6 months of age based on their association with later ASD-related behaviors provide an accurate prediction of ASD in an individual infant at 24 months.	96.6%
Guo et al., 2017 rs-fMRI [73]	ABIDE I	55	55	12.7 ± 2.4	76.4%	ROI-based FC matrices.	Deep neural networks with feature selection (DNN-FS)	Goal: Classification of ASD patients versus HCs. Conclusion: Identifying functional connections in brain regions associated with ASD, such as the default-mode network, cingulo–opercular network, frontal–parietal network, and cerebellum, can aid in ASD diagnosis.	86.36%
Jahedi A 2017 fcMRI [74]	ABIDE	126	126	17.3 ± 6.0	80.6%	ROI-based FC matrices.	Conditional random forest (CRF) RF	Goal: Identification of sensitive and specific biomarkers for ASD diagnosis. Conclusion: Combined RF and CRF achieved the best prediction accuracy.	92.7%
Kam et al., 2017 rs-fMRI [75]	ABIDE	119	144	<20	n.s.	Seed-based FC.	Discriminative restricted Boltzmann machine (DRBM)	Goal: Identification of dominant FCs to differentiate ASD from HCs. Conclusion: multiple clusters using hierarchical-level clustering of networks achieved high accuracy in discriminating ASD from HCs.	Single cluster 67.42% Multiple cluster 80.82%
Sadeghi et al., 2017 rs-fMRI [76]	ABIDE	31	29	20.49 ± 6.16	100%	ROI-based FC matrices.	SVM	Goal: Extract local and global parameters of brain networks to identify ASD. Conclusion: – SVM showed superiority to other classifiers. – Local parameters of the brain connectome in default mode, salience ventral attention, control, somatomotor and dorsal attention networks can be used for autism screening.	92%
Subbaraju et al., 2017 rs-fMRI [77]	ABIDE, PCP	505	530	6.5-58	84.8%	ROI-based FC matrices.	SVM	Goal: Spatial-feature-based detection method (SFM) extracts the most-discriminative BOLD signals. Conclusion: – Shift in resting state activities to prefrontal cortex in males. – SFM yielded accurate diagnosis of ASD.	78.6–95%
Tejwani et al., 2017 rs-fMRI [78]	ABIDE	147	146	n.s.	n.s.	ROI-based FC matrices.	SVM RF Naïve Bayes Multilayer perception algorithm	Goal: Identifying ASD versus HCs. Conclusion: – Increased FC in ASD more than HCs. – Dynamic FC measures are comparable with static FC measures such as node strength in predicting ASD.	65%
Heinsfeld et al., 2018 rs-fMRI [79]	ABIDE I	505	530	Site-specific	Site-specific	ROI-based FC matrices.	DNN	Goal: Classification of ASD and HCs. Conclusion: anterior–posterior underconnectivity in ASD.	70%
Bi et al., 2018 rs-fMRI [80]	ABIDE	45	39	13.4 ± 2.4	88%	ROI-based FC matrices.	Random SVM cluster	Goal: Classification of ASD and HCs using multiple SVMs. Conclusion: – Higher accuracy with random SVM. – Anomalies were also found in brain areas such as the inferior frontal gyrus, hippocampus, and precuneus.	96.15%
Fredo et al., 2018 rs-fMR [81]	ABIDE I, II	160	160	12.16 ± 2.76	100%	FC matrix	CRF	Goal: Classify ASD and HCs using CRF. Conclusion: – Using characteristics extracted from the cingulo–opercular task control (COTC) region, more precise classification can be achieved. – The connection between the COTC and the dorsal attention network was able to differentiate between individuals with ASD and healthy controls.	65%
Li et al., 2018 T-fMRI+rs-fMRI [82]	(T-fMRI) Clinical (rs-fMRI) ABIDE I	(T-fMRI) 82 (rs-fMRI) 41	(T-fMRI) 48 (rs-fMRI) 54	n.s.	n.s.	—	2-stage pipeline (DNN + prediction distribution analysis)	Goal: Detect brain region saliency in distinguishing ASD from HCs. Conclusion: The proposed method achieved efficient interpretation of deep learning with neuroimaging.	(T-fMRI) 87.1% (rs-fMRI) 85%
Bernas et al., 2018 rs-fMRI [83]	ABIDE	24	39	15.5 ± 1.0	87%	Wavelet coherence maps.	SVM	Goal: Extraction of temporal neurodynamic fMRI biomarkers for ASD diagnosis. Conclusion: Wavelet-coherence-based classifiers achieved robust and replicable results in ASD diagnosis.	86.7%
Bhaumik et al., 2018 rs-fMRI [84]	ABIDE	167	205	13.4 ± 5.1	81.7%	ROI-based FC.	SVM Partial least square regression (PLS)	Goal: Diagnosis and prediction of ASD by extracting neurobiological markers to differentiate ASD and HCs. Conclusion: Highest accuracy achieved with SVM and PLS.	62%
Dekhil et al., 2018 rs-fMRI [85]	NDAR	123	160	12.9 ± 3	53.3%	Power spectral densities	SVM	Goal: – To classify ASD and HCs. – Design a plan for personalized treatment. Conclusion: The proposed algorithm provided better diagnostic accuracy and prediction of clinical phenotypes.	91%
Xiao et al., 2018 fMRI [86]	ABIDE	42	42	9.78 ± 1.5	82.1%	Time courses of networks.	Stacked autoencoder (SAE) NN	Goal: Identification of school-aged ASD children from HC school-aged children. Conclusion: The accuracy and sensitivity can be improved by the use of all the frequency sub-bands.	87.2%
Yang et al., 2019 rs-fMRI [87]	ABIDE	505	530	6–6.4	84.8%	Time courses of networks.	SVM LR Ridge	Goal: classify ASD and HCs. Conclusion: combination of ML classifiers can improve the accuracy of ASD diagnosis.	71.98% with Ridge
Wang et al., 2019 (a) rs-fMRI [88]	ABIDE I, II	255	276	Site-specific	Site-specific	ROI-based FC.	SVM recursive features	Goal: To find the optimal features to achieve higher classification accuracy of ASD and HCs. Conclusion: High classification accuracy was achieved on both the global and the across-site datasets.	90.6%
Wang et al., 2019 (b) rs-fMRI [89]	ABIDE	501	533	n.s.	n.s.	ROI-based FC.	SVM recursive feature stacked sparse auto-encoder (SSAE)	Goal: To improve classification accuracy in identifying ASD and HCs. Conclusion: The proposed method can eliminate some meaningless features to enable the SSAE to extract insightful features.	93.59%
Aghdam et al. 2019 rs-fMRI [90]	ABIDE I, II	210	249	5–10	72.1%	Fast Fourier transformation	CNN	Goal: Diagnosis of ASD in young children. Conclusion: CNN is a powerful tool in the diagnosis of ASD in children.	70.5%
Huang et al., 2019 rs-fMRI [91]	ABIDE	45	47	11.1 ± 2.3	80.4%	Multiple group-sparse networks	SVM	Goal: To investigate how multiple networks can be used to generate different sparsity constraints and create a group of networks from a single brain. Conclusion: Better diagnosis was achieved by using multiple group-sparse FCNs.	79.4%
Jun et al., 2019 rs-fMRI [92]	ABIDE	121	171	14.4 ± 5.8	78.4%	Maps based on estimated likelihood of ROI by HMM	SVM	Goal: To distinguish between individuals with ASD and healthy controls, stochastic regional temporal BOLD fluctuations were analyzed, and their dynamic characteristics were estimated as likelihoods. Conclusion: The estimated likelihood of a regional BOLD signal from the corresponding region-wise HMM can be used to identify useful information for detecting ASD.	84.6%
Eslami et al., 2019 rs-fMRI [93]	ABIDE I	505	530	Site-specific	Site-specific	ROI-based FC	ASD-DiagNet	Goal: Classification of ASD and HC. Conclusion: High classification accuracy and AUC were achieved using ASD-DiagNet compared to other methods.	70.3%
Mostafa et al., 2019 rs-fMRI [94]	ABIDE I	403	468	Site-specific	Site-specific	The eigenvalues of the Laplacian matrix of brain network	LDA	Goal: Classification of ASD and HCs. Conclusion: Combining features of the eigenvalues of the Laplacian matrix of brain networks can help diagnose ASD more accurately.	77.7%
Song et al., 2019 rs-fMRI [95]	ABIDE	119	116	Site-specific	Site-specific	Community pattern analysis of FC	LDA	Goal: – Classification of ASD and HCs. – Prediction of the clinical class of ASD individuals. Conclusion: FC differences between ASD and HC revealed both under- and overconnectivity.	The highest accuracy achieved for on-site data was 85.16%. The maximum accuracy for multi-site data was around 75%.
Spera et al., 2019 rs-fMRI [96]	ABIDE	102	88	6.5–13	100%	ROI-based FC	Linear kernel SVM	Goal: Classification of ASD and HC by reducing the heterogeneity of age and sex. Conclusion: Mixed under- and overconnectivity patterns were found in the selected cohort of homogeneous age and sex.	71%
Tang et al., 2019 rs-fMRI [97]	ABIDE	42	37	n.s.	n.s.	ROI-based FC from DMN and whole brain	SVM Joint symmetrical non-negative matrix factorization (JSNMF)	Goal: Classification of ASD and HC. Conclusion: Better classification performance was obtained with training the classifiers with features extracted from DMN than the whole brain network.	AUC = 62.6
Yamagata et al., 2019 rs-fMRI [98]	Clinical	15	45	28.3 ± 6.1	100%	ROI-based FC	Sparse logistic regression (SLR) Multivariate ML approach	Goal: To distinguish pairs of ASD patients and their unaffected siblings from pairs of HCs and their siblings according to the endophenotype. Conclusion: – Multivariate ML approach can identify ASD endophenotype pattern of FC. – The endophenotype-related FCs are correlated with the clinical severity of ASD.	75%
Chaitra et al., 2020 fMRI [99]	ABIDE	432	556	n.s.	n.s.	ROI-based FC	Recursive-cluster-elimination SVM	Goal: Classification of ASD and HCs. Conclusion: Combined FC and graph measures achieved higher accuracy and better prediction than individual measures.	70.1%
Fan et al., 2020 [100]	ABIDE	145	157	16.4 ± 6.5	100%	Maps based on estimated likelihood values of ROI by HMM	SVM A hidden Markov model (HMM) was applied to two distinct groups in two distinct locations.	Goal: Classification of ASD and HC. Conclusion: Abnormalities in the frontopolar and orbitofrontal areas and the inferior temporal, middle temporal, amygdala, and fusiform gyri are prominent features of ASD, and they are associated with clinical functional deterioration.	74.9%
Liu et al., 2020 [101]	ABIDE	506	548	16.6 ± 8.1	85.3%	ROI-based FC	SVM elastic network method	Goal: – To diagnose autism spectrum disorder using an elastic network model based on data from resting-state functional magnetic resonance imaging (rs-fMRI) (ASD). – Aims to achieve an algorithm with high fitness and low model complexity by linearly adding a penalty term for estimated error and minimizing the residual sum of squares. Conclusion: – High classification accuracy in automatic diagnosis of ASD compared to LASSO and RR. – The elastic network method saves time and increases the algorithm’s effectiveness because it does not call for the pre-selection of features. Furthermore, compared to other algorithms, it has been shown to have better accuracy.	76.8%
Hu et al., 2020 [102]	ABIDE	403	468	n.s.	n.s.	ROI-based FC	Fully connected neural network (FCNN)	Goal: Classification of ASD and HC using FCNN and comparing results with other conventional classifiers. Conclusion: FCNN model achieved the highest classification accuracy.	69.8%
Sherkatghanad et al., 2020 [103]	ABIDE I	505	530	Site-specific	Site-specific	ROI-based FC	CNN	Goal: Classification of ASD and HCs using CNN. Conclusion: CNN model is computationally less intensive as it uses fewer parameters than state-of-the-art methods and can be used to prescreen ASD patients.	70.2%
Thomas et al., 2020 [104]	ABIDE I and II	620	542	5–64 Median = 13	80%	Nine summary measures	3D-CNN SVM	Goal: Classification of ASD and HCs. Conclusion: Comparable results of 3D-CNN to SVM algorithm were obtained, denoting that 3D CNN could not learn additional information in classifying ASD and HCs.	64%
Jiao et al., 2020 [105]	ABIDE I	505	530	n.s.	n.s.	ROI-based FC	CapsNET	Goal: – Distinguish ASD subjects from HCs. – Group people with ASD into various groups based on their unique functional connectivity metrics. Conclusion: – Comparing the CapsNET approach to other deep learning and ML techniques currently in use, the CapsNET approach showed superior classification results. – Heterogeneous FC patterns of ASD captured by CapsNet were consistent with existing neuropsychiatric findings and had a significant difference with ADOS score.	71%
Liu et al., 2020 [106]	ABIDE	250	218	Center specific	84.8%	ROI-based FC matrix	Elastic net	Goal: The elastic network model uses the rs-fMRI data to diagnose ASD. Conclusion: The elastic network approach saves time and increases algorithm efficiency because it does not call for feature selection in advance.	83.33%
Liu et al., 2020 [107]	ABIDE I	403	468	17.07 ± 7.95	83.5%	Dynamic FC	Multi-kernel SVM	Goal: To improve the classification accuracy of ASD and HCs by applying a multi-task feature selection method. Conclusion: The proposed method achieved higher accuracy compared to other multi-task methods.	76.8%
Zhang et al., 2020 [108]	ABIDE I	21	26	25.3 ± 6.3	100%	Approximate entropy (ApEn) Sample entropy (SampEn)	SVM	Goal: To investigate non-linear neural mechanisms that may be used as diagnostic biomarkers in people with ASD. Conclusion: The fast entropy method, which had lower entropy values and was quicker than conventional entropy methods, proved to be a more effective strategy for analyzing ASD patients than the FC method.	AUC = 62
Ronicko et al., 2020 [109]	ABIDE I, II	300	300	11.87 ± 2.8	80.5%	Partial and full correlation ROI-based FC matrix	RF SVM ORF NN	Goal: To classify ASD and HCs. Conclusion: In comparison to other strategies, the PCCE-CNN, PCCE-SVM, and MDMC-SVM methods all displayed superior performance.	70.3%
Khan et al., 2020 [110]	ABIDE	505	530	Site-specific	Site-specific	ROI-based FC matrix	SVM RF LR Decision tree Linear discriminant classifier	Goal: To detect ASD using a teacher–student-neural-network-based feature selection method. Conclusion: In 13 out of 17 site-specific comparisons, the suggested method outperformed other cutting-edge techniques in terms of accuracy.	82%
Reiter et al., 2021 [111]	ABIDE	306	350	6–18	n.s.	ROI-based FC matrix	RF	Goal: To investigate the impact of sample diversity on classification accuracy for ASD. Conclusion: Improved performance of RF classifier was found in the most-homogeneous samples.	73.75%
Devika, K., and Oruganti, V. R. M. 2021 [112]	ABIDE II	23	15	n.s.	84.2%	FC matrix	SVM	Goal: To detect ASD using ML algorithms. Conclusion: The proposed model showed effectiveness in classifying ASD and HCs.	80.76%
Ahammed et al., 2021 [113]	ABIDE I	19	19	15–35	78.9%	Bag-of-features extraction (BoF)	SVM	Goal: To classify ASD and HCs based on BoF extraction method. Conclusion: The use of BoF extraction method can support the clinical assessment and treatment of ASD.	81%
Ahammed et al., 2021 [114]	ABIDE I	79	105	15.25 ± 6.58	81%	ROI-based FC matrix of 3D-fMRI	DarkASDNet	Goal: To classify ASD and HCs. Conclusion: DarkASDNet provided a new benchmark method for classification of ASD individuals.	94.7%
Graña, M., and Silva, M. 2021 [115]	ABIDE	408	476	Site-specific	Site-specific	FC matrix	Nine classifiers	Goal: To explore the impact of choices during building up the ML pipelines on the predictive performance. Conclusion: Thorough analysis showed that using certain methods for feature extraction, such as factor analysis and principal component analysis (PCA), can improve the performance of the classifier (FA).	Best median AUC = 0.767
Al-Hiyali et al., 2021 [116]	ABIDE	41	41	n.s.	n.s.	Default mode network regions (DMN)	SVM K-nearest neighbors (KNN)	Goal: Utilized a variety of deep learning models to identify and categorize ASD using the temporal dynamic features of fMRI data. Conclusion: Deep models become more reliable and all-encompassing as more data from various sources are used to train them.	85.9% with KNN
Pominova et al., 2021 [117]	ABIDE II	184	168	10.15 ± 2.98	n.s.	FC matrices	3D convolutional autoencoders	Goal: To provide ASD recognition baselines based on FC matrices and full-size MRI series. Conclusion: – Eliminate site-related differences from fMRI. – Train robust neural network models that can be transferable between sites on a multi-site dataset.	—
Yin et al., 2021 [118]	ABIDE I	403	468	—	—	ROI-based FC matrix	Autoencoders CNN DNN	Goal: To distinguish ASD subjects from HCs based on graph theory and autoencoders. Conclusion: The proposed method of deep ML achieved higher accuracy than traditional ML algorithms.	79.2%
Chu et al., 2022 [119]	ABIDE	79	105	14.51 ± 6.23	79.9%	FC network regions	Multi-scale graph convolutional network (GCN)	Goal: Value of multi-scale graph representation learning (MGRL) framework in ASD diagnosis. Conclusion: Results demonstrated the effectiveness of MGRL in FCNs feature learning and ASD diagnosis.	0.795
Yang et al., 2022 [120]	ABIDE I	403	468	6–58	—	FC matrix	LR SVM DNN Supervised learning classifier	Goal: Distinguish ASD subjects from HCs based on FC metrics. Conclusion: For categorizing AIBDE fMRI data, the KSVM classifier is the best option.	69.43%

## Data Availability

Not applicable.

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
