# Peer review of "Role of Artificial Intelligence for Autism Diagnosis Using DTI and fMRI: A Survey"

_biomedicines, 2023, doi:10.3390/biomedicines11071858_

Round 1

Reviewer 1 Report

Your survey on the application of artificial intelligence (AI) for autism diagnosis using Diffusion Tensor Imaging (DTI) and functional MRI (fMRI) makes a substantial contribution to the understanding of current techniques and trends in this field. The comprehensive review you've performed is critical for improving early, objective detection and diagnosis of Autism Spectrum Disorder (ASD), an issue of significant societal importance.

Strengths:

·        The search methodology for your review is well-defined, including a broad array of keywords and thorough filtering of article types.

·        Your exploration of the application of AI and Machine Learning (ML) methodologies, particularly Deep Learning (DL) to ASD diagnosis, is timely and relevant given the current advances in these fields.

·        The discussion on emerging trends and potential improvements to current methodologies is useful for future research efforts in this field.

Suggestions for Improvement:

·        While you have gathered a wealth of information, it would be beneficial to include more specific details from the studies you reviewed. This could include specific results or insights that have been gleaned from these works.

·        A discussion on the practicality of using DTI and fMRI for ASD diagnosis in a clinical setting, including potential barriers to implementation such as cost and availability, would add value to your review.

·        Your conclusion presents an excellent overview of the potential of AI in ASD diagnosis. However, it could be strengthened by providing a more detailed summary of your key findings from the review.

Overall, your survey serves as an important reference for researchers and practitioners in the field of AI and ASD diagnosis. It highlights the potential of AI and machine learning techniques in transforming ASD diagnosis and provides valuable insights for further improvements.

Author Response

We would like to express our gratitude for the efforts of the reviewer for carefully reading and suggesting ways to improve our manuscript. The feedback was very helpful in improving the explanations and readability of the paper. Below is our point-to-point response.

  • While you have gathered a wealth of information, it would be beneficial to include more specific details from the studies you reviewed. This could include specific results or insights that have been gleaned from these works.

Reply: Thanks a lot for your suggestion. Insights that have been gleaned from the reviewed works have been added in the revised version. More specifically, the revised version includes three newly added figures that provide insights that have been gleaned from the reviewed works. More specifically, a summary of DTI and fMRI findings in autism is illustrated visually in Figure 1. The application of ML and AI to DTI and fMRI mainly involves two basic steps: feature extraction and classification. Figure 2 and Figure 3 visually summarize the most utilized features and classifiers in different reported DTI and fMRI autism diagnostic systems (please find the new Figures (Figure 1 on page 6, Figure 2 on page 9, and Figure 3 on page 15). A detailed discussion of the insights that have been gleaned from these works is detailed in the revised manuscript in sections 4.1 and 4.2, pages 24 and 25, lines 495-521.   

  • A discussion on the practicality of using DTI and fMRI for ASD diagnosis in a clinical setting, including potential barriers to implementation such as cost and availability, would add value to your review.

Reply: Thanks a lot for your suggestion. A discussion on the practicality of using DTI and fMRI for ASD diagnosis in a clinical setting, including potential barriers to implementation such as cost and availability, have been added to the review. Please, check the revised discussion section, lines 551-556, pages 25 and 26

  • Your conclusion presents an excellent overview of the potential of AI in ASD diagnosis. However, it could be strengthened by providing a more detailed summary of your key findings from the review.

Reply: Thanks a lot for your feedback. The conclusion section (section 5) has been extended to provide a more detailed summary of the key findings from the review. Please, check the updated conclusion section (page 26).

  • Overall, your survey serves as an important reference for researchers and practitioners in the field of AI and ASD diagnosis. It highlights the potential of AI and machine learning techniques in transforming ASD diagnosis and provides valuable insights for further improvements.

Reply: Thanks a lot for your positive feedback. We really appreciate your time reading and suggesting ways to improve the paper.

Reviewer 2 Report

The main objective of the paper is a literature review on applications of machine learning (ML) and artificial intelligence (AI) techniques to improve the diagnosis of autism using diffusion tensor imaging (DTI) and functional magnetic resonance imaging (fMRI). It contains a comprehensive summary of relevant literature in autism diagnosis with enlighten discussions, thus, the contribution of the paper is clear. In general, the literal presentation of the paper is good, explanations are understandable. However, I think that some image (DTI and fMRI) examples of principal results of the works reviewed and scheme(s) of the processing pipeline(s) showing the application of the ML and AI to the DTI and fMRI could improve the explanations and readability of the paper. In summary, I consider the contents of the paper are potentially publishable, but the minor issues commented above should be considered in a revised version of the paper.

Author Response

We would like to express our gratitude for the efforts of the reviewer for carefully reading and suggesting ways to improve our manuscript. The feedback was very helpful in improving the explanations and readability of the paper. Below is our point-to-point response.

The main objective of the paper is a literature review on applications of machine learning (ML) and artificial intelligence (AI) techniques to improve the diagnosis of autism using diffusion tensor imaging (DTI) and functional magnetic resonance imaging (fMRI). It contains a comprehensive summary of relevant literature in autism diagnosis with enlighten discussions, thus, the contribution of the paper is clear. In general, the literal presentation of the paper is good, explanations are understandable.

Reply: Thanks a lot for your positive feedback. We really appreciate your time reading and suggesting ways to improve the paper.

However, I think that some image (DTI and fMRI) examples of principal results of the works reviewed and scheme(s) of the processing pipeline(s) showing the application of the ML and AI to the DTI and fMRI could improve the explanations and readability of the paper.

Reply: Thanks a lot for your invaluable feedback which helps a lot in improving the explanations and readability of the paper. The revised version includes three newly added figures (Figure 1, Figure 2, and Figure 3) that provide examples of the principal results of the works reviewed and scheme(s) of the processing pipeline(s), showing the application of the ML and AI to the DTI and fMRI. More specifically, a summary of DTI and fMRI findings in autism is illustrated visually in Figure 1. The processing pipeline showing the application of ML and AI to DTI and fMRI mainly involves two basic steps: feature extraction and classification. Figure 2 and Figure 3 visually summarize the most utilized features and classifiers in different reported DTI and fMRI autism diagnostic systems (please find the new Figures (Figure 1 on page 6, Figure 2 on page 9, and Figure 3 on page 15). Moreover, a detailed discussion of these figures and their findings are detailed in the revised manuscript in sections 4.1 and 4.2, pages 24 and 25, lines 495-521.    

Round 2

Reviewer 1 Report

The authors have revised well and improved the quality and presentation.

I recommend to accept this manuscript for publication.

Author Response

Thanks for your positive feedback